# REFINING VISUAL REPRESENTATION FOR GENERALIZED ZERO-SHOT RECOGNITION THROUGH IMPLICIT-SEMANTICS-GUIDED METRIC LEARNING

## ABSTRACT

Deep metric learning (DML) is effective to address the large intra- and the small inter-class variation problem in visual recognition; however, when applied for generalized zero-shot learning (GZSL) in which the label of a target image may belong to an unseen category, this technique can be easily biased towards seen classes. Alternatively in GZSL some form of semantic space is available, which plays an important role in relating seen and unseen classes and is widely used to guide the learning of visual representation. To take advantage of DML while avoiding overfitting to seen classes, we propose a novel representation learning framework—Metric Learning with Implicit Semantics (MLIS)—to refine discriminative and generalizable visual features for GZSL. Specifically, we disentangle the effects of semantics on feature extractor and image classification of the model, so that semantics only participate in feature learning, and classification only uses the refined visual features. We further relax the visual-semantic alignment requirement, avoiding performing pair-wise comparisons between the image and the class embeddings. Experimental results demonstrate that the proposed MLIS framework bridges DML and GZSL. It achieves state-of-the-art performance, and is robust and flexible to the integration with several metric learning based loss functions.

## 1 INTRODUCTION

With the consideration of real-world recognition problems that may not have defined all classes during training, generalized zero-shot learning (GZSL) aims to leverage third-party data (e.g., attributes, semantic descriptors) to recognize samples from both of the seen and unseen classes Socher et al. (2013); Chao et al. (2016); Pourpanah et al. (2022). Therefore, a typical dataset for studying this problem is divided into two class sets: seen and unseen, with no intersection in between Xian et al. (2018a). Only samples of the seen classes are available for training the image recognition model; however, samples of both seen and unseen classes may appear during inference. The third-party data involving class-level semantic descriptors such as attributes are important in GZSL to relate seen and unseen classes. The knowledge learned from the seen classes must be generalized to recognize an unseen class through semantic information, because the visual data of unseen classes are absent in the training stage.

Whether a zero-shot setting is applied or not, an image recognition task can be categorized into *fine-grained* and *coarse-grained*, based on the amount of inter-class variation in visual appearance. Fine-grained recognition is considered more difficult than coarse-grained recognition due to subtle differences between classes. Nevertheless, the large intra-class variation in fine-grained recognition, often neglected in current studies, poses additional challenges to the task. Figure 1 displays a few samples in the CUB benchmark Wah et al. (2011). Some samples look quite differently to other samples in the same class. Such a large intra-class variation in appearance is inevitable because factors such as migration and molt may affect how birds change their colors.

Deep metric learning (DML) offers a natural solution to address large intra-class and small inter-class variance problem Hoffer & Ailon (2015); Wang & Chen (2017); Wang et al. (2019); Sun et al. (2020). It provides a flexible similarity measurement of data points. Each sample can have

a different penalty in updating the model. By optimizing the contrastive loss from positive pairs (intra-class) and negative pairs (inter-class), a model can leverage class-wise supervision to learn embeddings and give more penalties to hard samples. Furthermore, this technique can be applied on large scale dynamic and open-ended image datasets, and can allow extensions with limited efforts to new classes.

The semantic information in GZSL has been used to generate the visual features for unseen classes Xian et al. (2018b; 2019), as well as to guide the learning of discriminative visual features Ji et al. (2018); Zhu et al. (2019); Li & Yeh (2021). However, when DML is applied, a model can be easily overfitted to the seen classes despite the merits of DML mentioned above Bucher et al. (2016). Furthermore, a broad family of GZSL methods learn a joint embedding space, in which the classification is performed by directly comparing the embedded data points with the class prototypes Xian et al. (2018a). Learning such embedding functions can be difficult, because image features are extracted by a visual model pre-trained on ImageNet and class prototypes are human annotated attributes or are from word embeddings learned from text corpus. The visual and the semantic feature vectors may reflect inconsistent inter-class and intra-class discrepancies. The difficulty is exacerbated with the integration of generative methods to create visual features for unseen classes because the distribution of synthesized features is less predictable.

Delving into the image classification framework, it is composed of a feature extractor and a classifier Krizhevsky et al. (2012); He et al. (2016). Previous works typically use semantics in both feature extraction and classification Hu et al. (2020); Liu et al. (2021); Chen et al. (2021a); Chandhok & Balasubramanian (2021). As a result, the model is forced to align visual and semantic spaces, which may be difficult because of the modality gap mentioned above. Furthermore, semantics are used in both synthesizing visual features and learning embedding functions, which may introduce serious bias towards seen classes. To better leverage the semantic information, a viable solution is to refine visual embeddings by semantics, while the classification is performed only based on visual features.

Therefore, we present a novel representation learning framework, named Metric Learning with Implicit Semantics (MLIS), for GZSL. It takes advantage of metric learning to refine discriminative visual features from the original image features, while avoiding overfitting by making good (but not too extensive) use of semantics. MLIS decouples the effect of semantics on feature extractor and image classification, so that semantics only participate in feature learning, and classification only uses the refined visual features. This decoupling facilitates the training of both tasks. In feature learning we further relax the visual-semantic alignment requirement, avoiding performing pair-wise comparisons between the image and the class embeddings. To summarize, MLIS has the following characteristics that distinguish itself from existing methods:

- Semantic descriptors are given and fixed; they cannot be trained or fine-tuned. A GZSL model will reply on semantics to relate the seen and unseen classes; therefore fixing semantics reduces model complexity and thereby also reduces the chances of being overfitted.

- Semantic descriptors are involved only in training the encoder; *they are not used for downstream tasks (e.g., classification, segmentation)*. The downstream model utilizes only the visual features to perform the task. In this work semantic information is agnostic to the classification task.

- The entire framework learns only to refine visual features, and semantic descriptors *implicitly* affect the learning of visual features. We only pair an input visual feature vector up with its semantic descriptor to compute the loss, *not all semantic descriptors*.

- Visual-semantic alignment is not strictly enforced as we rely only on visual features to perform classification. The MLIS model is optimized to refine visual features so that when they are concatenated with the semantic descriptor of the target class, the metric learning based loss is minimized. We learn semantically meaningful visual features via metric learning from the semantics *without* aligning the visual and the semantic space.

We conduct extensive experiments on five benchmark datasets, including CUB Wah et al. (2011), AWA2 Xian et al. (2018a), SUN Patterson & Hays (2012), FLO Nilsback & Zisserman (2008), and aPY Farhadi et al. (2009). We demonstrate the superiority of the proposed method with performance on par with the state of the art. The code will be available upon paper acceptance.

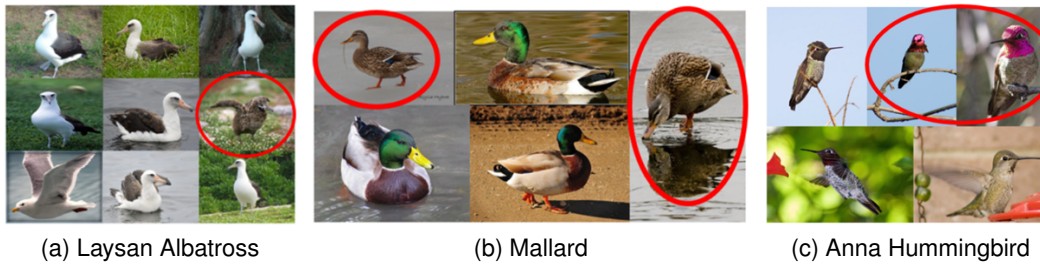

(a) Laysan Albatross    (b) Mallard    (c) Anna Hummingbird

Figure 1: Examples showing large intra-class variation in a fine-grained recognition task.

## 2    RELATED WORK

Zero-shot learning has evolved rapidly during the last decade and therefore documenting the extensive literature with limited pages is rarely possible. We review a few pioneering and recent zero-shot learning methods, with an emphasis on how the semantic information is utilized in these works.

A broad family of GZSL methods apply an embedding framework that learns a mapping between visual and semantic spaces. Some methods learned the mapping function from the visual space to the semantic space Akata et al. (2013); Frome et al. (2013); Socher et al. (2013). To alleviate the hubness problem caused by mapping from the high-dimensional visual space to the low-dimensional semantic space, some methods reversely learned the mapping from the semantic space to the visual space Changpinyo et al. (2016); Atzmon & Chechik (2019). The strategy of learning the mappings from these spaces to a common space has also been explored in Wang & Chen (2017). All of the embedding strategies perform the direct alignment of the visual and the semantic space, which can be difficult because of the inherent gap between them.

Recent methods fused the visual and the semantic features using concatenation Hu et al. (2020); Liu et al. (2021). For example, the FREE model Chen et al. (2021a) incorporated semantic to visual mapping into a unified generative model to refine the visual features. Both of the real and the fake visual features were used to reconstruct semantic embeddings and were converted into a low-dimensional latent space. The visual features concatenated with latent embedding, together with reconstructed pseudo-semantics, were used as inputs to train a classifier. In two-level adversarial visual-semantic coupling (TACO) Chandhok & Balasubramanian (2021), two generative models were applied to create visual and semantic features, both of which were used to train the classifier. Our approach is similar in that it uses concatenation of visual and semantic features. However, semantics are not involved in learning the target task in our method. Decoupling the effect of semantics on feature extractor and the downstream task is one of the main difference of our method to these works. Moreover, these approaches feed the concatenated features to many layers for computation, which in our view contradicts the goal of avoiding direct visual-semantic alignment.

Semantics have also been used to guide the learning of visual attention maps. For example, in Dense Attention Zero-shot LEaring (DAZLE) Huynh & Elhamifar (2020), different image regions were considered to correspond to different attributes, and therefore a dense attention mechanism was proposed to obtain attribute-based features from a local image region for each attribute. Semantic-guided multi-attention (SGMA) Zhu et al. (2019) also discovered the most discriminative parts of objects, which jointly learned cooperative global and local features from the whole object as well as the detected parts to categorize objects based on semantic descriptions. The stacked semantics-guided attention ($S^2GA$) model Ji et al. (2018) obtained semantic relevant features by using individual class semantic features to progressively guide the visual features to generate an attention map for weighting the importance of different local regions. TCN Jiang et al. (2019) performed an element-wise product operation on visual and semantic features to determine whether image and semantic information are consistent. These methods align the visual and the semantic features harder through an attention mechanism and learn the visual features based on seen classes. Therefore, the methods may easily be overfitted to seen classes, leading to inferior generalization ability on unseen classes.

Finally, semantics are also utilized to be part of the input to generative models to alleviate the extreme data-imbalance problem by synthesizing unseen class features. The synthesized features

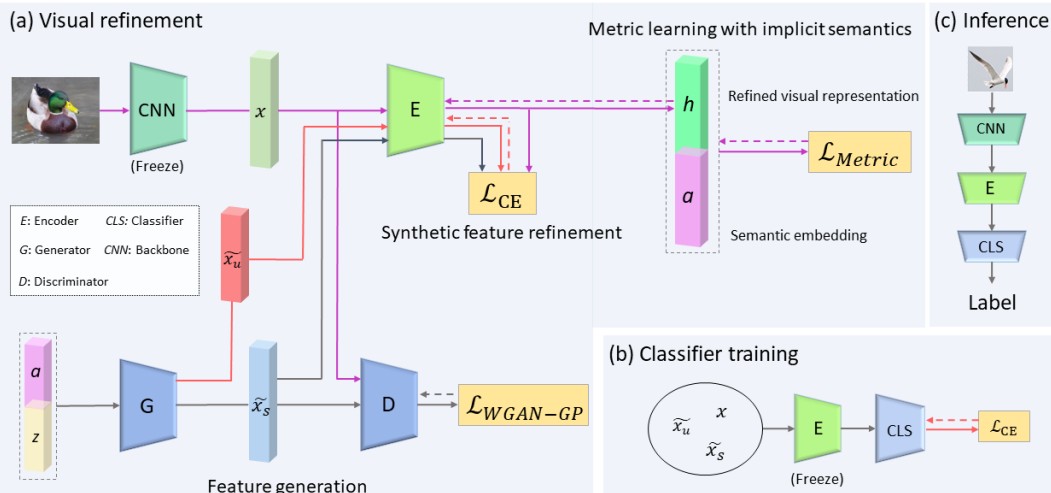

Figure 2: Overview of the proposed MLIS framework. (a) **Visual refinement**: this stage learns visual representations. An encoder is trained to extract discriminative and generalizable visual features. It receives both of the real and the synthetic data. (b) **Classifier training**: when the encoder is trained until converge, we train a classifier based on visual features alone. (c) **Inference**: the inference is performed by using the encoder and the classifier trained in the previous stages.

can be fed to conventional classifiers so that GZSL is converted into the conventional supervised learning problem Long et al. (2017). Generative models such as variational autoencoders Bucher et al. (2017); Verma et al. (2018); Schönfeld et al. (2019); Ma & Hu (2020), Generative Adversarial Networks (GANs) Chen et al. (2018); Li et al. (2019) and their variants Xian et al. (2019) have been used. Following the trend, the proposed method also contains a generative model to synthesize unseen class features. Recently, the generative models are used to create not only visual features but also semantic features Chen et al. (2021a); Chandhok & Balasubramanian (2021). Our approach does not allow changes of semantic space to reduce the chance of overfitting.

Our approach has some similarities with CE-GZSL Han et al. (2021). This method learns two visual embeddings (class-level and instance-level) from an image, while ours learns a single one. Their class-level contrastive embedding method concatenates the visual embedding with *all* semantic embeddings—the visual embedding of an instance is directly affected by all semantic descriptors. We pair up only with the corresponding semantic descriptor—the visual embedding of an instance is indirectly affected by other semantic descriptors and the model complexity is significantly reduced.

In summary, existing methods alter the semantic space Chen et al. (2021a); Chandhok & Balasubramanian (2021), align hardly the visual and the semantic spaces Huynh & Elhamifar (2020); Zhu et al. (2019); Ji et al. (2018); Jiang et al. (2019), and let semantics participate in learning downstream models Chen et al. (2021a); Chandhok & Balasubramanian (2021); Hu et al. (2020); Liu et al. (2021). We believe that those characteristics hinder the development of a powerful GZSL model. Generating fake semantic features alters the original semantic space and makes the representation learning task more difficult. Strictly aligning the visual and the semantic spaces is difficult to achieve and the model must handle the gap between them. Involving semantics in downstream tasks makes the model less focused on the visual cues and also increases the chance of overfitting to seen classes. The proposed MLIS method strives to escape the negative influence caused by these factors.

## 3 METHOD

The overall architecture is displayed in Fig. 2, consisting of a pre-trained Convolutional Neural Network (CNN), an encoder (E) and a classifier (CLS). Besides, it contains a generator (G) and a discriminator (D) to create visual features for seen and unseen classes. The entire procedure is divided into three stages: visual refinement, classifier training and inference. The visual refinement

stage (Fig. 2(a)) focuses on refining discriminative and generalized visual representations to facilitate classifier learning. Through the implicit guidance of semantics specified in this method, the visual representation learning task exploits the advantages of metric learning that can accurately define the notion of similarity between samples. In the classifier training stage (Fig. 2(b)), semantics will *not* participate in this process, which can not only alleviate the learning difficulty of the classifier but also improve the training efficiency. After the first two stages are accomplished, the inference is trivial using the trained encoder and the classifier (Fig. 2(c)). We focus on the visual refinement stage and provide technical details in the following subsections.

### 3.1 METRIC LEARNING WITH IMPLICIT SEMANTICS

To deal with the large intra-class variation problem, in which hard samples can be very far from the optimal solution, we apply metric learning to manipulate the similarity relationship between samples. In metric learning each sample can have its own weight to influence model training in backward propagation based on the similarity to other samples. This process is similar to that of mining because it finds the best samples to train on.

In order to put more penalty on the loss of hard samples so that they can move towards the correct feature cluster further in each update, we apply the circle loss Sun et al. (2020) as the loss function to optimize the models. In circle loss, different similarity scores should have different penalties. This loss aims to simultaneously maximize the within-class similarity $s_p$ and minimize the between-class similarity $s_n$. The loss is defined as follows:

$$L_{circle} = \log[1 + \sum_{j=1}^{L} \exp(\gamma(s_n^j + m)) \sum_{i=1}^{K} \exp(\gamma(-s_p^i))], \tag{1}$$

where the scale factor $\gamma$ and the relaxation margin $m$ are two hyperparameters of this loss. We will empirically analyze their effects in Section 4.5. $L$ and $K$ are the numbers of between-class and within-class similarity scores, which may vary in different batches depending on the training instances sampled in a batch. The similarity between samples is calculated by the cosine similarity.

However samples of unseen classes are not available during training in GZSL. In the absence of training data for unseen classes, metric learning can be easily overfitted to the seen classes, leading to the lack of generalization capability on unseen classes. Therefore, we propose Metric Learning with Implicit Semantics (MLIS), considering both of the semantic and the visual features in calculating the circle loss. Specifically, a training sample is represented by a concatenated feature vector consisting of the refined visual features (outputted by $E$) and the semantic embedding of the target class (given, fixed). Note that one class contains only a semantic feature vector. Therefore, images of the same class may have different visual features, but are concatenated with a fixed semantic feature vector to calculate the sample similarity to other samples. Note that we do not compare a visual embedding $h$ and a semantic embedding $a$. Instead, we concatenate them. The dimension of the visual and the semantic embedding can be different.

This indirect alignment between visual and semantic information is more flexible and achievable because the refined visual embeddings do not require to align perfectly to the semantic embeddings. We seek for refined visual features, when concatenated with corresponding semantic features, can be classified well. The semantics act as prior to guide the visual representation learning. Furthermore, although a visual feature vector is only concatenated with the ground truth semantic feature vector, the representation learning is affected by the *entire* semantic distribution. The encoder must consider inter-class similarities between semantic embeddings in learning the visual representations. Suppose the encoder does nothing and returns the input vector (original features); in this extreme case, the classification is difficult because of the gap between two spaces. Now, suppose the encoder does a perfect alignment: a sample is represented by its semantic descriptor, leading to an extremely easy classification task (samples of the same class has one unique representation; each class has its own representation). The tradeoff between representation learning and classification is considered in MLIS, enabling indirect visual-semantic alignment.

As will be demonstrated in Sec. 4.4, we have experimented with the circle loss and a few other metric learning based loss functions in MLIS. All of them have improved performance, especially on the unseen classes.

### 3.2 FEATURE GENERATION

To compensate the lack of training samples for unseen classes, a generator (G) is trained in the visual refinement stage to generate visual features for unseen classes. It can generate more features for seen classes as well. Therefore, the classifier can be trained with samples of both seen and unseen classes in the next stage.

We develop our generative model based on WGAN-GP Gulrajani et al. (2017), which is easier to train than Generative Adversarial Networks (GANs) Goodfellow et al. (2014). In particular, the gradient penalty (GP) has been shown in Gulrajani et al. (2017) to be more effective than the weight clipping used in Wasserstein GAN (WGAN) Arjovsky et al. (2017). We follow Xian et al. (2018b) to extend the improved WGAN Gulrajani et al. (2017) to a conditional WGAN by integrating the semantic embeddings. The loss defined in WGAN-GP is:

$$L_{\text{WGAN-GP}} = \mathbb{E}[D(\tilde{x}, a)] - \mathbb{E}[D(x, a)] + \lambda \mathbb{E}[(\|\nabla_{\hat{x}} D(\hat{x}, a)\|_2 - 1)^2], \tag{2}$$

where $\tilde{x} = G(z, a)$, $\hat{x} = \alpha x + (1 - \alpha) \tilde{x}$ with $\alpha \sim U(0, 1)$, $\lambda$ is the penalty coefficient, $a$ is a semantic descriptor, and $z$ is a random noise.

### 3.3 SYNTHETIC FEATURE REFINEMENT

Now we pay attention to the input features to our encoder. The encoder is trained by minimizing the circle loss Sun et al. (2020), which helps extract an effective visual representation to differentiate between seen classes. We do not compute the similarity between real and synthesized features, because those supervisory signals are not reliable. To improve the quality of the synthesized features ($\tilde{x}_u$ and $\tilde{x}_s$), we add a classification loss to further improve the discrimination between both of the real and the fake features.

Therefore, the encoder is optimized by using two losses: the circle loss considers semantic embeddings and uses different penalty to improve the discrimination between seen classes and to ensure the stability of the model, while the classification loss is used to separate all classes (including seen and unseen and involving real and synthesized features). The conventional cross-entropy is used in this work:

$$L_{\text{CE}} = -\frac{1}{N} \sum_i \sum_{c=1}^{M} y_{ic} \log(p_{ic}), \tag{3}$$

where $M$ is the number of classes, $y_{ic}$ indicates whether the $i$-th sample belongs to the class $c$, and $p_{ic}$ is the predicted probability.

### 3.4 CLASSIFIER TRAINING AND INFERENCE

Finally we describe the classifier training stage (Fig. 2(b)) and the inference stage (Fig. 2(c)). To train the image classifier, we use not only real features but also synthesized features (for both seen and unseen classes). The features are further refined by using the encoder. Finally, we use the refined features of all samples to train a softmax classifier as the final GZSL classifier. During inference, the pre-trained CNN model, the encoder, and the classifier are used to predict the class label of a test image. Note that the classification process involves only visual features.

## 4 EXPERIMENTS

### 4.1 DATASETS AND EVALUATION PROTOCOL

We evaluated the proposed method on five GZSL benchmark datasets, including CUB Wah et al. (2011), AWA2 Xian et al. (2018a), SUN Patterson & Hays (2012), FLO Nilsback & Zisserman (2008), and aPY Farhadi et al. (2009). We followed the experimental settings provided by Xian et al. (2018a) and summarized them in Table 1. We calculated the top-1 accuracy of seen classes ($S$) and unseen classes ($U$), and reported the harmonic mean: $H = \frac{2*U*S}{U+S}$. Each experiment was run five times and the mean accuracy scores were reported.

Table 1: Statistics of the datasets used in our experiments. (*tr*: training set, *ts*: test set, *S*: seen classes, *U*: unseen classes)

| Dataset | Number of classes | | | Number of images | | |
|---------|-------|------|--------|-----------|-----------|-----------|
| | Total | Seen | Unseen | $\mathbf{Y}_S^{tr}$ | $\mathbf{Y}_S^{ts}$ | $\mathbf{Y}_U^{ts}$ |
| CUB Wah et al. (2011) | 200 | 150 | 50 | 7,057 | 1,764 | 2,967 |
| AWA2 Xian et al. (2018a) | 50 | 40 | 10 | 23,527 | 5,882 | 7,913 |
| SUN Patterson & Hays (2012) | 717 | 645 | 72 | 10,320 | 2,580 | 1,440 |
| FLO Nilsback & Zisserman (2008) | 102 | 82 | 20 | 5,631 | 1,403 | 1,155 |
| aPY Farhadi et al. (2009) | 32 | 20 | 12 | 5,932 | 1,483 | 7,924 |

## 4.2 IMPLEMENTATION DETAILS

We used the visual features provided by Xian et al. (2018a) in the experiments. They are 2048-dimensional features extracted by ImageNet-1K Krizhevsky et al. (2012) pre-trained ResNet-101 He et al. (2016). For semantic features, we used category-level attributes for AWA2, SUN, and aPY. For CUB and FLO, we followed Narayan et al. (2020); Xian et al. (2019); Jiang et al. (2019); Li et al. (2019); Xian et al. (2018b) and used 1024-dimensional semantic descriptors extracted from textual descriptions of the images by character-based CNN-RNN Reed et al. (2016).

The encoder contains one fully connected (fc) layer with ReLU. The classifier contains one fc layer with logSoftmax. The generator and the discriminator are both two fc layers with LeakyReLU. The metric learning loss functions were developed based upon Musgrave et al. (2020).

Table 2: Comparison with state-of-the-art methods. $U$ and $S$ are the top-1 accuracy of unseen and seen classes. $H$ is the harmonic mean of $U$ and $S$. The best result is marked in red, and the second best result is marked in blue.

| Method | Venue | CUB | | | AWA2 | | | SUN | | | FLO | | | aPY | | |
|--------|-------|-----|-----|-----|-----|-----|-----|-----|-----|-----|-----|-----|-----|-----|-----|-----|
| | | U | S | H | U | S | H | U | S | H | U | S | H | U | S | H |
| TCN | ICCV'19 | 52.6 | 52.0 | 52.3 | 61.2 | 65.8 | 63.4 | 31.2 | 37.3 | 34.0 | - | - | - | 24.1 | 64.0 | 35.1 |
| LisGAN | CVPR'19 | 46.5 | 57.9 | 51.6 | 52.6 | 76.3 | 62.3 | 42.6 | 36.6 | 39.4 | 57.7 | 83.8 | 68.3 | 34.3 | 68.2 | 45.7 |
| f-VAEGAN-D2 | CVPR'19 | 48.4 | 60.1 | 53.6 | 57.6 | 70.6 | 63.5 | 45.1 | 38.0 | 41.3 | 56.8 | 74.9 | 64.6 | - | - | - |
| DAZLE | CVPR'20 | 56.7 | 59.6 | 58.1 | 60.3 | 75.7 | 67.1 | 52.3 | 24.3 | 33.2 | - | - | - | - | - | - |
| DVBE | CVPR'20 | 53.2 | 60.2 | 56.5 | 63.6 | 70.8 | 67.0 | 45.0 | 37.2 | 40.7 | - | - | - | 32.6 | 58.3 | 41.8 |
| Keshari *et al.* | CVPR'20 | 44.8 | 59.9 | 51.3 | 59.5 | 73.4 | 65.7 | 44.8 | 42.9 | 43.8 | - | - | - | - | - | - |
| ZIC-LDM | Sensors'21 | 40.3 | 62.9 | 49.1 | 31.9 | 92.5 | 47.4 | 23.5 | 33.9 | 27.6 | - | - | - | - | - | - |
| TACO-GZSL | WACV'21 | 61.2 | 57.7 | 59.4 | 59.4 | 74.2 | 66.0 | 44.0 | 39.0 | 41.3 | 60.6 | 81.1 | 69.4 | - | - | - |
| Li *et al.* | AAAI'21 | 51.1 | 58.2 | 54.4 | 56.9 | 80.2 | 66.6 | 36.6 | 47.6 | 41.4 | - | - | - | - | - | - |
| CN-GZSL | ICLR'21 | 49.9 | 50.7 | 50.3 | 60.2 | 77.1 | 67.6 | - | - | - | - | - | - | - | - | - |
| HSVA | NIPS'21 | 52.7 | 58.3 | 55.3 | 56.7 | 79.8 | 66.3 | - | - | - | - | - | - | - | - | - |
| GCM-CF | CVPR'21 | 61.0 | 59.7 | 60.3 | 60.4 | 75.1 | 67.0 | 47.9 | 37.8 | 42.2 | - | - | - | 37.1 | 56.8 | 44.9 |
| FREE | ICCV'21 | 55.7 | 59.9 | 57.7 | 60.4 | 75.4 | 67.1 | - | - | - | 67.4 | 84.5 | 75.0 | - | - | - |
| VGSE-SMO | CVPR'22 | 24.1 | 45.7 | 31.5 | 45.7 | 66.7 | 54.2 | 25.5 | 35.7 | 29.8 | - | - | - | - | - | - |
| TDCSS | CVPR'22 | 44.2 | 62.8 | 51.9 | 59.2 | 74.9 | 66.1 | - | - | - | 54.1 | 85.1 | 66.2 | - | - | - |
| MLIS-GZSL | Ours | 66.1 | 66.7 | 66.4 | 62.0 | 78.2 | 69.2 | 45.6 | 37.9 | 41.4 | 62.8 | 82.9 | 71.5 | 34.6 | 66.1 | 45.4 |

## 4.3 COMPARISON WITH STATE-OF-THE-ARTS

First, we compare the recognition performance of the proposed method with state-of-the-art methods including Jiang et al. (2019); Li et al. (2019); Xian et al. (2019); Huynh & Elhamifar (2020); Min et al. (2020); Keshari et al. (2020); Liu et al. (2021); Chandhok & Balasubramanian (2021); Li et al. (2021); Skorokhodov & Elhoseiny; Chen et al. (2021b); Yue et al. (2021); Chen et al. (2021a); Xu et al. (2022); Feng et al. (2022). Table 2 shows the comparison results.

The proposed method achieve the best results in CUB and AWA2. In the CUB experiment, our performance is 4-5% higher in accuracy than the second best performance for both seen and unseen classes, and is 6.1% higher in h-mean. In the AWA2 experiment, we also have the best h-mean accuracy and have a more consistent performance on recognizing seen and unseen classes. We achieve the second best results in FLO and aPY in all evaluation metrics. The accuracy on SUN is not as good as those on other datasets. We observe that the accuracy rates of all methods on SUN

are relatively low. One possible reason is that this dataset is composed of scene categories such as "arena rodeo", "arena soccer", "art school", "art studio" rather than specific objects. Moreover, it has more classes (717 classes) than other datasets and has fewer images (only 20) per category.

The proposed MLIS framework re-investigates the role of semantics in GZSL. Although the training set contains samples only from the seen categories, our encoder can extract discriminative visual features that are also generalizable to unseen classes because we maintain the original semantic space. The learned visual representations are implicitly affected by the semantic embeddings as well as the relationship between them. The classification is performed solely based on visual features; therefore, image features are not required to be matched with their semantic features, avoiding the requirement of performing direct visual-semantic alignment. However, the learned visual embedding space is still semantically meaningful, so that the classification performance in this space is satisfactory.

## 4.4 ABLATION EXPERIMENTS

We investigate the effect of the critical components introduced in the proposed framework. This ablation experiment involves the comparison of methods including (1) the direct application of Metric Learning (ML)—circle loss Sun et al. (2020) in this experiment—on representation learning, (2) the proposed metric learning with Implicit Semantic (IS), and (3) the addition of synthetic feature refinement (SFR), in which a classification loss is further considered to improve the discrimination of visual representations between seen and unseen classes. Table 3 shows the results.

Table 3: Ablation study on effects of components in the proposed framework

| ML | IS | SFR | CUB | | | AWA2 | | | SUN | | | FLO | | | aPY | | |
|---|---|---|---|---|---|---|---|---|---|---|---|---|---|---|---|---|---|
| | | | U | S | H | U | S | H | U | S | H | U | S | H | U | S | H |
| ✓ | ✗ | ✗ | 54.7 | 45.0 | 49.3 | 53.7 | 73.6 | 62.1 | 46.0 | 36.1 | 40.5 | 61.8 | 81.1 | 70.1 | 29.9 | 60.9 | 40.1 |
| ✓ | ✓ | ✗ | 65.0 | **67.0** | 66.0 | **62.6** | 78.0 | 69.4 | 45.6 | 36.2 | 40.4 | 60.8 | 79.8 | 69.0 | 35.1 | 62.2 | 44.9 |
| ✓ | ✓ | ✓ | **66.6** | 66.7 | **66.6** | **62.6** | **79.7** | **70.1** | **47.3** | **38.0** | **42.2** | **62.1** | **85.6** | **72.0** | **35.2** | **68.0** | **46.4** |

We can observe that using metric learning alone does not achieve state-of-the-art performance on all datasets. For example, the performance in CUB, AWA2, and aPY are not satisfactory. With the proposed implicit semantics, the performance on those datasets can be significantly boosted, as displayed in the second row of Table 3. The proposed IS method is designed to be compatible with a metric learning based loss, taking advantage of how it learns discriminative visual features. Finally, SFR can be used to further improve the discrimination between seen and unseen samples. This component helps improve the performance in all datasets. The results validate that metric learning with implicit semantics (IS) achieves a better discrimination between samples than using metric learning alone. Adding SFR can achieve a stable performance of the seen (or the unseen) class and improve the accuracy of the other.

Table 4: Performance of MLIS with three different metric learning based loss functions

| Loss | IS | CUB | | | AWA2 | | |
|---|---|---|---|---|---|---|---|
| | | U | S | H | U | S | H |
| Circle loss Sun et al. | ✗ | 54.7 | 45.0 | 49.3 | 53.7 | 73.6 | 62.1 |
| (2020) | ✓ | **65.0** | **67.0** | **66.0** | **62.6** | **78.0** | **69.4** |
| Contrastive loss | ✗ | 46.4 | 51.2 | 48.7 | **54.0** | 73.3 | 62.2 |
| Hadsell et al. (2006) | ✓ | **58.7** | **62.5** | **60.5** | 53.8 | **76.9** | **63.3** |
| Multi-similarity loss | ✗ | 51.3 | 25.2 | 33.8 | 45.7 | 77.0 | 57.4 |
| Wang et al. (2019) | ✓ | **65.1** | **67.9** | **66.5** | **56.5** | **80.4** | **66.4** |

To further demonstrate that MLIS can improve other metric learning based methods, we replaced the circle loss with a few different loss functions including contrastive loss Hadsell et al. (2006) and multi-similarity loss Wang et al. (2019). As the goal of this experiment was not to compare the performance of these loss functions, we did not fine-tune the hyperparameters for each of them. We reported the experimental results on CUB (fine-grained) and AWA2 (coarse-grained). Table 4

shows the results. With the appropriate usage of semantics, all of the metric learning methods can be significantly boosted and are less biased toward seen classes.

## 4.5 HYPERPARAMETER ANALYSIS

In this section, we analyzed the effect of the number of synthesized features on the performance. In the MLIS framework, we can synthesize features in the visual refinement stage and the classifier training stage. Figure 3 shows the results on the CUB dataset (fine-grained) and the AWA2 dataset (coarse-grained), in which Fig. 3(a)(b) shows the analysis results in the visual refinement stage and Fig. 3(c)(d) shows those in the classifier training stage. In the visual refinement stage, a relatively great performance was obtained when the generator created five samples for each unseen class. Synthesizing more samples, somehow surprisingly, decreased the accuracy of unseen classes. However, in the classifier training stage, we needed a sufficient amount of synthesized features for each unseen class to achieve good performances. This is explainable because we rely solely on visual cues to perform classification.

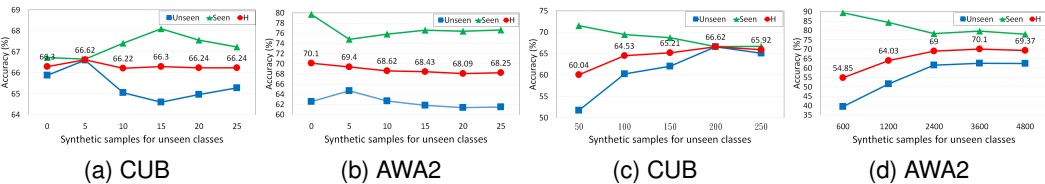

Figure 3: Effect of the number of synthesized samples to the classification performance. (a)-(b):the visual refinement stage, (c)-(d): the classifier training stage.

Finally, we analyzed the effect of the two hyperparameters $m$ and $\gamma$ in circle loss on the classification performance. The relaxation factor $m$ controls the radius of the circular decision boundary, and the scale factor $\gamma$ determines the largest scale of each similarity score. Figure 4(a)(b) show the effect of the hyperparameter $m$ on CUB and AWA2. A small $m$ value (below 0.4) is favorable, which is consistent with the finding reported in the original paper Sun et al. (2020). Figure 4(c)(d) show the effect of the hyperparameter $\gamma$ on CUB and AWA2. The classification performance is fairly stable with respect to this parameter.

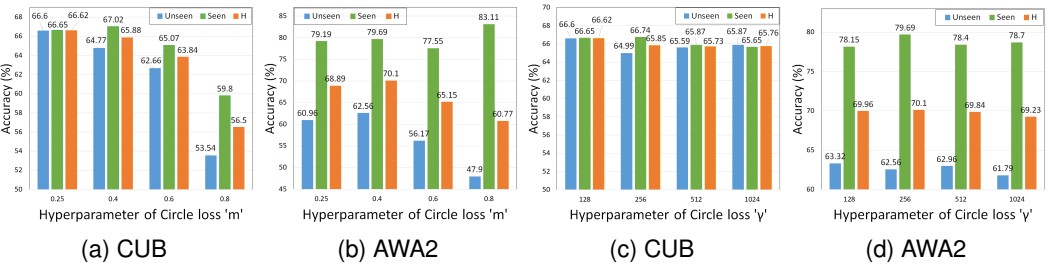

Figure 4: Effect of the hyperparameters $m$ and $\gamma$ in circle loss to the classification performance.

## 5 CONCLUSION

In this paper, we re-investigate the role of semantics in the generalized zero-shot learning task. We identify and address two major issues: overfitting to seen classes and large intra-class variation. We propose MLIS—a visual representation framework to shine metric learning in the generalized zero-shot setting. The MLIS method disentangles the effects of semantics on feature extractor and image classification of the model, and prevents the direct alignment of the visual and the semantic spaces. In the experiments, we demonstrate the superiority of the MLIS method with performance on par with the state of the art on five benchmarks.

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
