# OpenReview forum: "Refining Visual Representation for Generalized Zero-Shot Recognition through Implicit-Semantics-Guided Metric Learning"
_ICLR.cc/2023/Conference — Submitted to ICLR 2023_

### Official Review · Reviewer_WJ1v · 2022-10-24

**Confidence:** 4
**Correctness:** 3
**Technical Novelty And Significance:** 2
**Empirical Novelty And Significance:** 2
**Recommendation:** 3

**Clarity, Quality, Novelty And Reproducibility:**

Clarity: the paper is written clearly and is easy to understand.
Quality & Novelty: As I mentioned above, I feel the paper lacks novelty.
Reproducibility: The paper gives clear instructions on the implementation. However, it lacks enough descriptions on its training schema, which makes the experiments hard to reproduce only relying on the paper.

**Strength And Weaknesses:**

Strength:
1. The paper is written clearly and easy to follow
2. The proposed method is extensively evaluated on multiple benchmark datasets and shows competitive results.

Weakness:
1. This paper lacks novelty. Using metric learning to improve the feature quality and generative model to synthesize unseen features are well studied in the GZSL domain. The essential techniques used in this paper, including the WGAN-GP for feature generation, and the circle loss, are proposed in previous works.
2. The empirical performance is not compelling enough to me, considering the method is mainly composed of existing techniques.
3. It lacks some qualitative results to justify the motivation. The paper was motivated by the fact that fine-grained GZSL classification is more sensitive to intra-class variations. So it is better to have some visualization (like tSNE graphs) to demonstrate that the refined features have fewer intra-class divergences compared to the raw features and other methods.

**Summary Of The Paper:**

This paper proposes a new method, metric learning with implicit semantics (MLIS), for generalization zero-shot learning (GZSL). It leverages a metric learning loss during training to refine the visual feature and better align the semantic information. In addition, to alleviate the feature bias towards the seen classes, it also uses a generative model to simulate the unseen class features to improve the overall quality. The proposed MLIS was evaluated on multiple benchmark datasets for GZSL, and competitive performance has been observed.

**Summary Of The Review:**

This paper proposes an interesting method that can improve the feature quality for GZSL through metric learning and feature synthesis. However, it does not have enough novelty, and the performance margin is not large enough to justify an acceptance.

---

### Official Review · Reviewer_tWPT · 2022-10-24

**Confidence:** 5
**Correctness:** 3
**Technical Novelty And Significance:** 2
**Empirical Novelty And Significance:** 3
**Recommendation:** 3

**Clarity, Quality, Novelty And Reproducibility:**

The paper is well written and easy to follow.
The contributions are incremental.


**Strength And Weaknesses:**

Strengths
1.	This paper is well written and easy to follow.
2.	The proposed framework is simple and effective, and easy to reproduce.

Weaknesses
1.	The motivation is unclear. The authors consider that semantics used in both synthesizing visual features and learning embedding functions will introduce bias toward seen classes. However, some methods [1][2] using semantics seem to get better results. Please compare with them and give more explanations for the motivation of the proposed paper. Moreover, it would be more convincing to add comparative experiments that use semantics but do not decouple.
2.	The comparison on the results of the CUB is unfair. Most of the methods, such as FREE and f-VAEGAN-D2, utilize 312-dimensional attributes as the auxiliary semantic information. You need to experiment with attributes rather than 1024-dimensional semantic descriptors if you want to compare with these methods.
3.	Some recent methods like [2] and [3] are ignored to be compared.
4.	I wonder why the results are so low using only ML in the ablation experiments. The results are even lower than some simple early methods like f-CLSWGAN [4] and f-VAEGAN-D2 [5]. More explanations can be given.
5.	A minor problem. In section 3.4, the authors said that synthesized features including both seen and unseen classes are used to train the final classifier. However, it seems that only the synthesized unseen features are used.

[1] Generative Dual Adversarial Network for Generalized Zero-shot Learning. CVPR 2019.
[2] Latent Embedding Feedback and Discriminative Features for Zero-Shot Classification. ECCV 2020.
[3] Contrastive Embedding for Generalized Zero-Shot Learning. CVPR 2021.
[4] Feature Generating Networks for Zero-Shot Learning. CVPR 2018.
[5] F-VAEGAN-D2: A feature generating framework for any-shot learning. CVPR 2019.


**Summary Of The Paper:**

This paper introduces deep metric learning for generalized zero-shot learning, and proposes a novel framework named MLIS to avoid overfitting to seen classes. MLIS disentangles the effect of semantics on feature learning and classification, making the training of the two tasks more effective. Moreover, this paper concatenates semantic descriptors and visual features directly instead of aligning them, reducing the model’s complexity. Experimental results demonstrate the effectiveness of the proposed MLIS.

**Summary Of The Review:**

This paper is clear and easy to follow. However, the contributions still need to be explained more clearly. A fair comparison is also needed on the CUB dataset.

---

### Official Review · Reviewer_wRXx · 2022-10-25

**Confidence:** 5
**Clarity, Quality, Novelty And Reproducibility:** Limited novelty and far below the sta…
**Correctness:** 2
**Technical Novelty And Significance:** 2
**Empirical Novelty And Significance:** 2
**Recommendation:** 3

**Strength And Weaknesses:**

The organization is good. Nevertheless，


1.The motivation for the usage of metric learning for improving the generalization is not new. In fact, using metric learning to solve ZSL have been explored in previous works (e.g., Yannick Le Cacheux et al. In ICCV19 ). The overall compared methods are limited, and more methods should be surveyed and compared.

2.The framework is actually incremental improvements on existing methods, e.g., improved on Yongqin Xian et al. in CVPR 2018 by feeding additional unseen features as inputs. From this point, the novelty is limited to the community, since this kind of strategies have been explored in the community (e.g., in LsrGAN of ECCV20). Furthermore, the idea of disentangling feature extraction and classifier training is not new. All these aspects reduced the contributions of this paper to the community.

3.The overall writing is poor. The formulas are also confusing. Please amend them accordingly.

4.How to initialize the network weights, from scratch? What’s the specific network architecture?

5.Since additional operation such as two-step decoupled training is used for the proposed method, I am doubted about the tradeoff between the running time and accuracy.

6.The parameter analysis is shown, however, what’s the specific parameters for achieving these results? It seems the authors have choosed to compare with the methods worse than theirs.

**Summary Of The Paper:**

This paper proposes Metric Learning with Implicit Semantics (MLIS) to refine discriminative and generalizable visual features for GZSL. MLIS disentangles the effects of semantics on feature extractor and image classification of the model, so that semantics only participate in feature learning, and classification only uses the refined visual features. MLIS relaxes the visual-semantic alignment requirement, avoiding performing pair-wise comparisons between the image and the class embeddings.Experiments are conducted on golden GZSL datasets.

**Summary Of The Review:**

Due to limited novelty, I must reject it!

---

### Official Review · Reviewer_iKnN · 2022-11-02

**Confidence:** 4
**Correctness:** 3
**Technical Novelty And Significance:** 2
**Empirical Novelty And Significance:** 2
**Recommendation:** 3

**Clarity, Quality, Novelty And Reproducibility:**

The overall clarity of this paper is fine, but as is discussed above, some parts need improvement.

I am not saying this paper is novel. Currently, it seems like a simple combination of GAN and circle loss. (See weakness above)

ZSL experiments are usually easy to reproduce as datasets are quite small. However, in this paper, a lot of implementation details are not given. Such as the hyper-parameter of the circle loss, the number of intra-class and inter-class pairs and the weights of the losses in training. I am also afraid of the performance on CUB as usually, GAN-based ZSL models are not able to produce such a high H score on it.



**Strength And Weaknesses:**

## Pros
---
* Clear expression of the method
* Good reference to the related articles (though it seems to be too much)

## Cons
---
* This paper has several unclear or confusing parts. Some may be due to sloppy writing, but some show a lack of in-depth thinking into the problem.
  * In Fig. 2, the MLIS loss is termed as $L_{Metric}$, but in the main body, it becomes $L_{circle}$.
  * $L_{circle}$ is based on cosine similarity. So what about $L_\text{CE}$? There is no clue of how the class-wise probability $p$ is obtained.
  * The authors need to clarify whether the generated unseen samples are involved in computing $L_{circle}$ and the values of $L$ and $K$. This is important as otherwise the generative model would not be related with $L_{circle}$ and then could be removed from the entire framework.
* $L_{circle}$ is built upon concatenated features ($h$ and $a$). Usually, they need to be calibrated before concatenation. Uncalibrated features could lead to problems in computing cosine similarity, including
  * Bad magnitude normalizer
  * Un-normalized similarity bias provided by $a$
* $L_{circle}$ and $L_\text{CE}$ are functionally overlapping, but their presence is different, i.e., circle loss vs cross-entropy. I think the authors need to explain this inconsistency in loss design.
* The overall design is not very novel to me. GAN + classification loss just looks like InfoGAN with some add-ons, while the circle loss is also off-the-shelf. The concept of MLIS does not bring anything new as it is just a simple modification on the input side of the circle loss. This does not convince me to be a good ICLR paper.
* Please be aware of the font difference in subscriptions between $L_{circle}$ and $L_\text{CE}$.
* Marginal performance gain

**Summary Of The Paper:**

This paper talks about ZSL. The authors ground their motivation on the fact that in-class samples could be still very different from each other, and they propose to tackle this problem in ZSL. The main contribution yields the concept of Metric Learning with Implicit Semantics (MLIS) and it is plugged into the training stage of a generative ZSL model.

The proposed model is basically extending a GAN in ZSL. MLIS classifies the visual samples by a circle loss when the generative model is being trained. An additional cross-entropy loss is also applied. A default GAN loss counts the last part of the model's training objective.

**Summary Of The Review:**

I would say the quality of this paper is below the expectation of a regular ICLR paper. It lacks novelty, and some designs are not properly explained.

---

### Decision · Program_Chairs · 2023-01-20

**Decision:**

Reject

**Justification For Why Not Higher Score:**

No reviewers recommend acceptance.

**Justification For Why Not Lower Score:**

N/A

**Metareview: Summary, Strengths And Weaknesses:**

This paper proposes to cope with the bias towards seen classes in zero-shot learning, by decoupling the use of semantics in feature learning vs classification. While the proposed method is simple and evaluation is extensive, the proposed method is incremental, parts of the experimental setup could be improved, and the writing could be clearer.